# Endogenous production of hyaluronan, PRG4, and cytokines is sensitive to cyclic loading in synoviocytes

**Meghana Pendyala**[1,2], **Paige S. Woods**[3], **Douglas K. Brubaker**[4,5], **Elizabeth A. Blaber**[1,2,6], **Tannin A. Schmidt**[3], **Deva D. Chan**[1,2,4]*

**1** Department of Biomedical Engineering, Rensselaer Polytechnic Institute, Troy, New York, United States of America, **2** Center for Biotechnology and Interdisciplinary Studies, Rensselaer Polytechnic Institute, Troy, New York, United States of America, **3** Department of Biomedical Engineering, University of Connecticut Health Center, Farmington, Connecticut, United States of America, **4** Weldon School of Biomedical Engineering, Purdue University, West Lafayette, Indiana, United States of America, **5** Regenstrief Center for Healthcare Engineering, Purdue University, West Lafayette, Indiana, United States of America, **6** Blue Marble Space Institute of Science at NASA Ames Research Center, Moffett Field, California, United States of America

* chand@purdue.edu

**Data Availability Statement:** All confocal image files and raw data files are available from the Harvard Dataverse database (https://doi.org/10.7910/DVN/GFLGBK).

## Abstract

Synovial fluid is composed of hyaluronan and proteoglycan-4 (PRG4 or lubricin), which work synergistically to maintain joint lubrication. In diseases like osteoarthritis, hyaluronan and PRG4 concentrations can be altered, resulting in lowered synovial fluid viscosity, and pro-inflammatory cytokine concentrations within the synovial fluid increase. Synovial fibroblasts within the synovium are responsible for contributing to synovial fluid and can be targeted to improve endogenous production of hyaluronan and PRG4 and to alter the cytokine profile. We cyclically loaded SW982 synoviocytes to 0%, 5%, 10%, or 20% strain for three hours at 1 Hz. To assess the impact of substrate stiffness, we compared the 0% strain group to cells grown on tissue culture plastic. We measured the expression of hyaluronan turnover genes, hyaluronan localization within the cell layer, hyaluronan concentration, PRG4 concentration, and the cytokine profile within the media. Our results show that the addition of cyclic loading increased *HAS3* expression, but not in a magnitude-dependent response. Hyaluronidase expression was impacted by strain magnitude, which is exemplified by the decrease in hyaluronan concentration due to cyclic loading. We also show that PRG4 concentration is increased at 5% strain, while higher strain magnitude decreases overall PRG4 concentration. Finally, 10% and 20% strain show a distinct, more pro-inflammatory cytokine profile when compared to the unloaded group. Multivariate analysis showed distinct separation between certain strain groups in being able to predict strain group, hyaluronan concentration, and PRG4 concentration from gene expression or cytokine concentration data, highlighting the complexity of the system. Overall, this study shows that cyclic loading can be used tool to modulate the endogenous production of hyaluronan, PRG4, and cytokines from synovial fibroblasts.

**Funding:** MP and DDC received funding from the National Science Foundation (Awards #1944394 and #2149946, https://www.nsf.gov) for this work. The funders had no role in study design, data collection and analysis, decision to publish, or preparation of the manuscript.

**Competing interests:** The authors have declared that no competing interests exist.

## Introduction

Intraarticular synovial fluid is a viscous solution that works to reduce friction between the articulating surfaces of the joint. Synovial fluid also is a pool for nutrient transport and regulatory cytokines [1]. The synovial membrane that surrounds the joint consists of synovial fibroblasts (or synoviocytes), which contribute to synovial fluid, and macrophages/monocytes within a collagenous extracellular matrix [2]. Synovial fluid contains plasma proteins [3, 4], but its lubricating ability is due largely to hyaluronan, secreted by synoviocytes, and PRG4, secreted by both chondrocytes and synoviocytes [5, 6]. While in synovial fluid in healthy joints is composed of high molecular weight hyaluronan and high concentrations of PRG4 [7, 8], osteoarthritic (OA) synovial fluid has decreased hyaluronan molecular weight and may affect PRG4 concentrations within the synovial fluid, increased hyaluronan fragments and cytokine concentrations [7, 9–11]. Increasing hyaluronan concentrations in synovial fluid exogenously have been used as treatments for deteriorating joint health, but the benefits are short-lived [12].

Hyaluronan concentration and molecular weight is regulated by a balance of synthesis and degradative activity. Hyaluronan is synthesized by three membrane-bound proteins (HAS1, -2, -3), each of which produce hyaluronan of different molecular weight distributions and localization and are active under different contexts. HAS3 produces hyaluronan of a smaller size (between $1 \times 10^5$ and $1 \times 10^6$ kDa) than HAS1 or HAS2 (between $2 \times 10^5$ and $2 \times 10^6$ kDa) [13]. While HAS2 produces the majority of hyaluronan in healthy joints, HAS1 has been shown to be more involved in synovitis during OA [14]. Conversely, degradation enzymes, like hyaluronidases, break down hyaluronan chains into fragments. Expression of hyaluronidase 2 increased during OA, promoting breakdown of hyaluronan and shifting the molecular weight distribution lower [15]. Increasing expression of *HAS2* has been shown to increase hyaluronan concentrations [16, 17], which makes targeting endogenous hyaluronan production an ideal mechanism for a long-term solution for OA.

Likewise, PRG4 production is reduced with OA, highlighting the need to preserve both functional components of the synovial fluid. PRG4 works in a dose-dependent manner to lower friction within the joint [18, 19]. PRG4 deficiency has been shown to diminish the lubricating ability of synovial fluid, while PRG4 supplementation has been shown to improve the boundary lubrication function of synovial fluid [9, 20].

Synoviocytes also modulate the inflammatory environment of the synovium. Acting as effector cells, they produce pro-inflammatory cytokines, proteases, and affect synovial macrophage phenotype [21, 22]. The resulting cytokine concentrations within the synovial fluid can act as biomarkers to determine OA severity [23]. IL-6, VEGF, MMP-1, eotaxin, CCL3, and CCL2 levels have been shown to be higher in synovial fluid from patients with OA when compared to healthy controls [11, 24, 25]. Conversely, a reduction in pro-inflammatory cytokines, such as TNFα or IL-1β, via targeted therapies has been shown to alleviate pain in patients with OA [26].

Despite observations of changes to hyaluronan, PRG4, and cytokine content within the synovium with disease, it remains unclear how to modulate these synoviocyte products within a mechanically active joint. While studies that observe the effect of mechanical stimulation on synovial fluid components are limited, cyclic strain has been shown to be a promising mechanism by which hyaluronan and PRG4 synthesis can be increased. Exercise has been shown to improve proteoglycan and collagen content in cartilage [27–29] and decrease the severity of OA [30]. Cyclic compression has been shown to alter the expression of hyaluronan synthases in synoviocytes embedded within a collagen gel [31] and to alter the expression of inflammatory and matrix molecules [32–35], supporting the protective potential of mechanical stimulus

on other tissues in the joint. These studies, however, have not measured changes in hyaluronan degradation or production of other components of synovial fluid, like PRG4. Mechanical stimulation of tissues also affects cytokine regulation, such as in dermal fibroblasts [36], alveolar epithelial cells [37] and synovial fibroblasts [38, 39]. Determining how mechanical stimulation can promote hyaluronan and PRG4 production and modulate cytokine production from synovial fibroblasts will enhance understanding of the role the synovium plays in maintaining a healthy joint environment.

Therefore, the objective of this study was to determine the extent to which short-term mechanical stimulation affects endogenous hyaluronan, PRG4, and cytokine production in synovial fibroblasts, across a range of strain magnitudes. We hypothesized that synoviocytes will increase production of hyaluronan and PRG4 and reduce pro-inflammatory cytokine production with higher strain magnitudes.

## Methods

### Cell culture

Human synovial sarcoma fibroblasts (SW982, ATCC, Manassas, VA, Cat: HTB-93™) were maintained in complete medium: high glucose (4.5 g/L) Dulbecco's modified Eagle's medium (DMEM) supplemented with 10% fetal bovine serum, 1% 100-units/mL penicillin, 1% 10-μg/mL streptomycin, 110 mg/L sodium pyruvate, and 1% 2-mM L-glutamine. All cell culture media and supplements were sourced from VWR (Randor, PA).

### Cyclic loading regimen

SW982 cells were seeded on silicone membranes (BioFlex plates, Flexcell International Corp, Burlington, NC) at a density of $3-4 \times 10^4$ cells/cm$^2$ (optimized during pilot experiments to minimize cell detachment while preserving adequate total RNA yield) and were allowed to attach for 24 hours in 1 mL complete medium. Cells were confined to the central area of the membrane by using plastic molds (BioFlex Cell Seeders, Flexcell International Corp) underneath each well and applying vacuum pressure, creating a designated area for cell seeding. Cells were subjected to cyclic equibiaxial strain to a peak magnitude of 0%, 5%, 10%, and 20% ($n = 4$) and a frequency of 1 Hz (sinusoidal waveform) using the FX6000T Tension System (Flexcell International Corp). Load was applied for three hours by stretching the flexible membrane over loading posts using a controlled vacuum (Fig 1A). Cells were also cultured on tissue culture plastic (TCP) as a control. To evaluate the early gene expression changes in response to loading, media was removed from 3 wells and cell layers were lysed and pooled for RNA isolation as described below, four hours after the end of loading. Prior work has demonstrated that changes to hyaluronan synthase expression can occur as early as immediately after external stimulation [40, 41]. 24h after the end of loading, media was collected from the remaining wells and stored at -80˚C in aliquots (Fig 1B). Cell layers were fixed for immunofluorescence staining.

### Gene expression

RNA was isolated from cell layers using the PureLink RNA Mini Kit (Invitrogen, Waltham, MA) following manufacturer's instructions. Cell layers were washed with PBS after media removal and lysed using the provided lysis buffer and were homogenized using a 23g needle, passing through a 1 mL syringe 8–10 times, and deposited into an RNase free microcentrifuge tube. An equal volume of 70% ethanol was added to the tube and the contents were vortexed and transferred into a spin column for washes. RNA was eluted out using 30 uL of RNase free

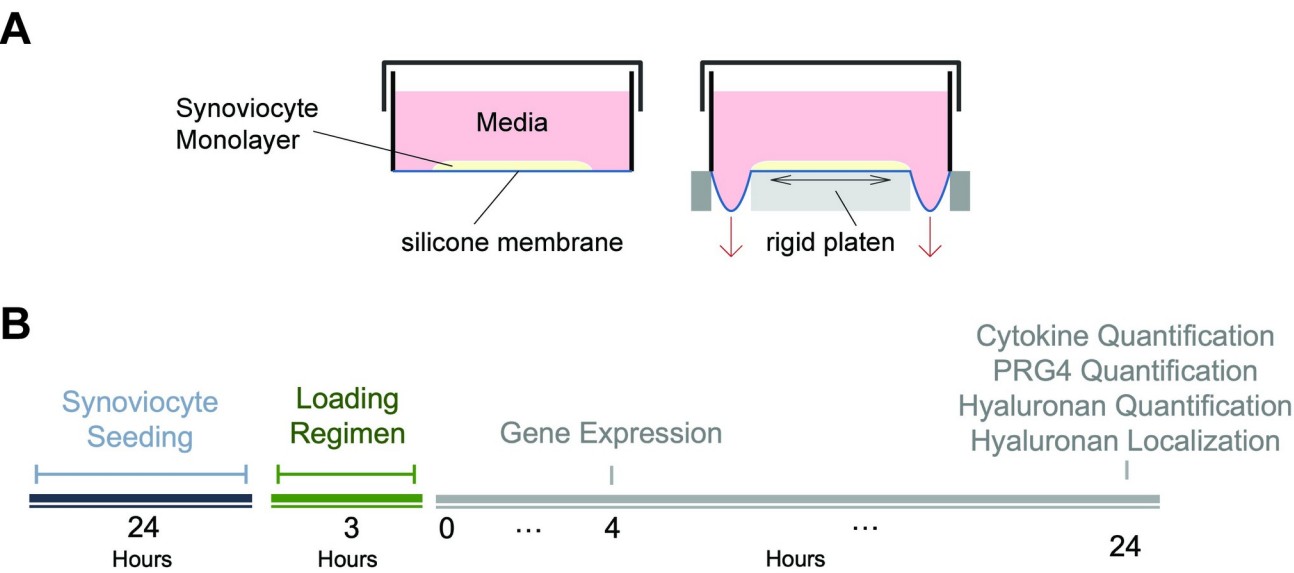

**Fig 1. Experimental set-up of Flexcell loading system.** A) Vacuum-controlled pressure is used to stretch the silicone membrane evenly across the loading post. B) Synoviocytes were seeded for 24 hours and loaded for 3 hours before measuring outcomes at specific timepoints.

water and quality was checked using the Synergy H4 Hybrid Micropate Reader with a Take 3 cartridge (BioTek). cDNA was synthesized using SuperScript IV, and qPCR (3 technical replicates) was performed using manufacturer validated TaqMan® primers (Invitrogen, listed in S1 Table) to measure *HAS1* (hyaluronan synthase 1), *HAS2* (hyaluronan synthase 2), *HAS3* (hyaluronan synthase 3), *HYAL1* (hyaluronidase 1), *HYAL2* (hyaluronidase 2), *CEMIP* (cell migration inducing hyaluronidase), and *TMEM* (cell surface hyaluronidase). GAPDH, which showed minimal variation in Cq values among all groups (S2 Table), was used as the housekeeping gene. Relative transcript abundance was calculated as $2^{-\Delta Cq}$, where $\Delta Cq$ is the difference between Cq of the gene of interest and Cq of *GAPDH*. To assess the extent to which synoviocyte response was due to change in substrate stiffness, relative transcript abundance was compared between the TCP and 0% strain groups. Fold difference was then calculated for Flexcell-seeded groups as $2^{-\Delta\Delta Cq}$, where $\Delta\Delta Cq$ is the difference between $\Delta Cq$ of the experimental group and $\Delta Cq$ of 0% strain, to evaluate the strain-magnitude dependent response.

### Hyaluronan concentration quantification

We determined hyaluronan concentration in the media samples at 4 and 24 hours after loading using a previously published method [42]. 1 mL of media was dialyzed against 0.1 M NaCl overnight at room temperature using the 3.5kDa Diaeasy Dialyzer tubes (ThermoFisher Scientific, Waltham, MA). Protein in the samples were digested using 25 µL of 19-mg/mL proteinase K in 10 mM Tris at 60˚C overnight. Proteinase activity was stopped by heating samples to 100˚C for 15 min. Samples were then dialyzed against deionized water overnight at room temperature before measuring hyaluronan content in triplicates with a commercially available competitive ELISA-like assay (#K-1200, Echelon Biosciences, Salt Lake City, UT) read using a multiplate reader (Biotek Synergy H4). A standard curve was generated using a four-parameter curve following manufacturer's protocol to extrapolate the hyaluronan concentration of sample. Percent difference was calculated against the 0% strain group.

## Immunofluorescence

Immunofluorescence staining was used 24 hours after the end of the loading to localize hyaluronan within the cell layers. Cell monolayers were fixed with ice cold methanol for 10 minutes and washed with PBS. Cells were stained for hyaluronan following a previously published method [43]. Cells were incubated with 5 mg/mL biotinylated hyaluronan binding protein (Sigma Aldrich, St. Louis, MO) at 4°C for 16 hours. After washing, the cells were incubated with streptavidin Alexa Fluor 488 (Jackson ImmunoResearch Laboratories, West Grove, PA) at a 1:500 dilution for 1 hour at room temperature. After another wash, cells were blocked for non-specific binding using 1% BSA in PBS on ice for 15 minutes and then incubated with 0.2% Triton-X 100 on ice for 15 minutes. Cells were incubated with Phalloidin Atto 565 (Millipore Sigma) to stain for F-Actin for 40 minutes at room temperature After a final wash, the silicon membrane of the plates was cut out and mounted onto glass slides with coverslips using Vectashield with DAPI (Vector Laboratories, Burlingame, CA).

Slides were imaged on a Zeiss LSM 800 (Zeiss Microscopy, Jena, Germany) confocal microscope using the Zeiss Sen Blue Edition imaging software. Z-stacks were acquired at three separate locations on each slide and maximum intensities were projects from each stack. The resulting files were processed using FIJI/ImageJ [44, 45] to obtain fluorescent intensities of hyaluronan and actin staining [46]. Hyaluronan fluorescent intensity was normalized to F-Actin fluorescent intensity, using integrated densities of each stain subtracted by the background fluorescence after a rolling ball background subtraction at a 50.0-pixel radius.

## PRG4 quantification

Conditioned media samples from 24 hours after the completion of loading were analyzed for PRG4 concentration using the AlphaLISA platform (PerkenElmer), as described previously [47]. Briefly, streptavidin modified donor beads were bound to biotin-labeled recombinant human PRG4 (Lubris BioPharma, Naples, FL) [48], and anti-PRG4 mAb 94D6 was bound to acceptor beads. Competitive inhibition of the rhPRG4 –mAb 4D6 interaction by free unlabeled PRG4 in solution was used for PRG4 quantification. Percent difference was calculated against the 0% strain group.

## Cytokine quantification

Conditioned media samples were probed for cytokine concentrations using the Bio-Plex Pro Human Cytokine 27-plex Assay (Bio-Rad Laboratories, Hercules, CA) following manufacturer's protocol. Absorbance values were read using a BioPlex 3D Luminex system (Bio-Rad Laboratories). Standard curves were generated for each cytokine using 5-parameter logistic curves, which were used to determine concentrations for each sample replicate. Out of range concentrations were below the lower limit or above the upper limit of the standard curve. Cytokines whose concentrations fell below the lower limit for all samples and were excluded from analyses. Percent difference for loaded groups was calculated against the 0% strain group.

## Statistical analysis

Statistical testing was performed using R [49] in RStudio [50], with statistical significance defined at $\alpha = 0.05$ for all hypothesis tests. Shapiro-Wilk's test was used to confirm normality for each experimental outcome. Levene's and Bartlett's tests were used to compare sample variances in normal or non-normal datasets, respectively. The effect of strain magnitude (0%, 5%, 10%, and 20% strain) was tested using one-way analysis of variance (ANOVA) or a Kruskal-

Wallis test for non-normal datasets. Any significant effects were tested for differences among groups using a Tukey's honest significant post-hoc or a Dunn's test. Differences between the TCP and 0% strain were probed using an unpaired student's t test (normal datasets with equal variances), unpaired Welch's t test (normal datasets with unequal variances), or a Mann-Whitney rank test (non-normal datasets). Many groups had a non-normal distribution for $\Delta$Cq and $\Delta\Delta$Cq data, so all gene expression data was evaluated with non-parametric tests. Multivariate analysis was performed using R (mixOmics package [51]). Sparse principal component analysis (sPCA) and sparse partial least squares (sPLS) was performed to predict strain group, hyaluronan concentration, and PRG4 concentration from gene expression and cytokine concentration. Sparse analysis uses a regularization procedure to subset data into tighter clusters with redundant variables removed.

## Results

To assess the synoviocyte response to a change in substrate stiffness and 5%, 10%, and 20% cyclic strain at 1 Hz, we used qPCR to measure changes in expression of hyaluronan synthesis and breakdown genes, ELISAs to measure hyaluronan and PRG4 concentrations, and a multiplex panel to measure cytokine concentrations in media.

### Effect of change in substrate stiffness on gene expression

*HAS1*, *HAS2*, and *HAS3* gene expression was not significantly affected by substrate stiffness (Fig 2, S3 Table). However, *HYAL1* (p = 0.029, effective fold difference vs. TCP: -5.00 ± 0.045) expression was significantly reduced with culture on silicone membranes, while *HYAL2 CEMIP*, and *TMEM* were not significantly impacted by the difference in substrate.

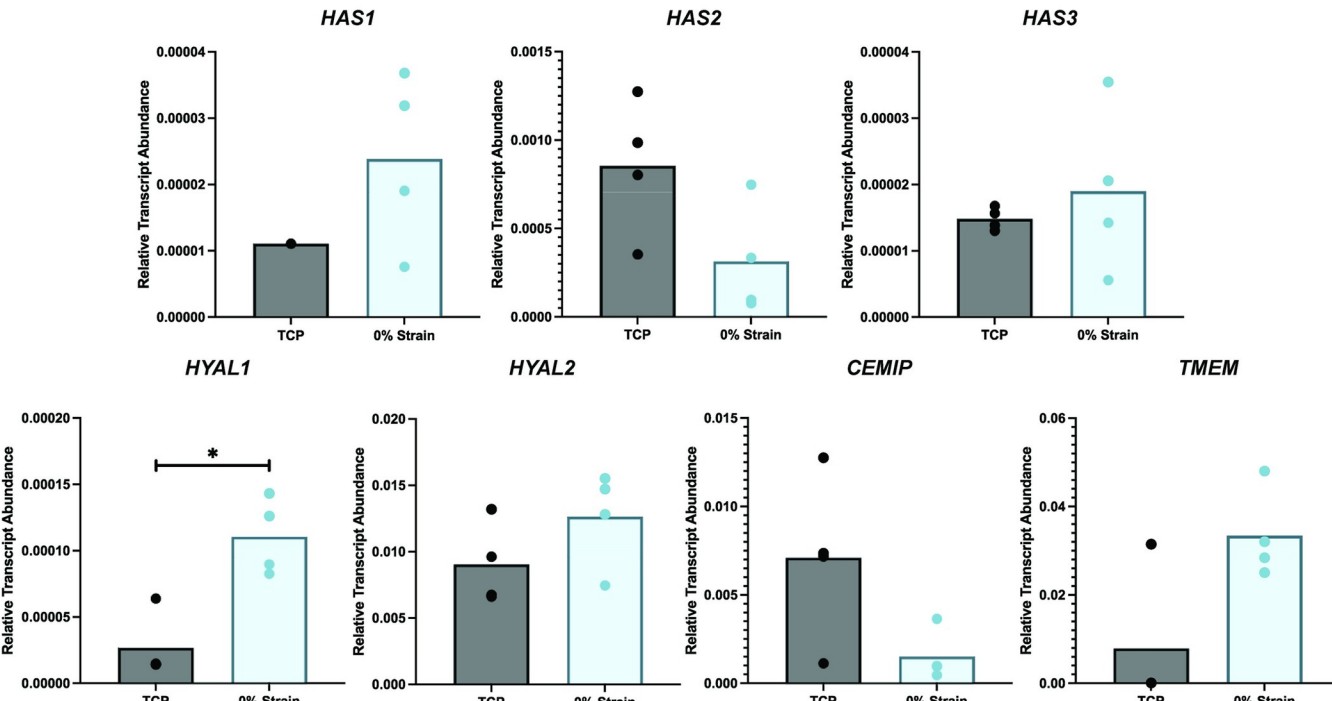

**Fig 2. Change in material stiffness affects hyaluronidase gene expression, but not synthase expression.** Relative transcript expression was calculated from $\Delta$Cq values, with *GAPDH* as a housekeeping gene, as $2^{-\Delta Cq}$. Dots represent individual samples and significant differences (*, p < 0.05) between groups are shown with brackets.

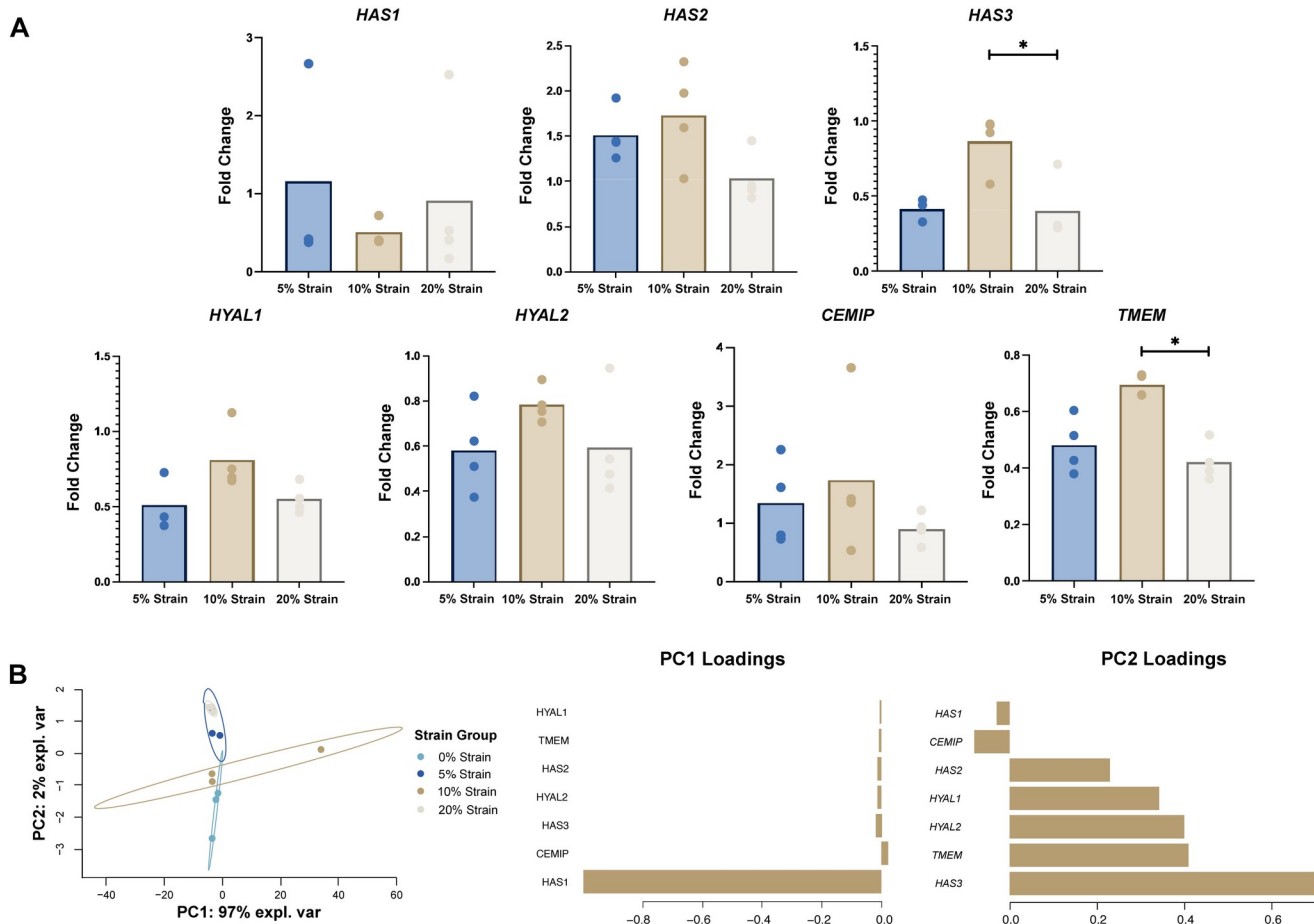

**Fig 3. Changes in strain magnitude have limited effects on hyaluronan turnover genes compared to 0%-strain conditions.** A) Mean fold difference in gene expression of loaded samples was calculated with respect to 0% strain group. Dots represent replicates and significant difference (*, p < 0.05) between groups are shown with brackets. B) sparse PCA of gene expression clustered by strain group, with associated loadings plots. sPCA is shown as a scatter plot in the X-Y representative space of the gene expression data along PC1 and PC2, which explain over 99% of the variation in the data.

## Effect of different strain magnitudes on gene expression

*HAS1* and *HAS2* expression was not significantly different among the 0%, 5%, 10%, and 20% strain groups (Fig 3, S1 Fig, S3 Table). *HAS3* expression was significantly different among groups (p = 0.041). All loaded groups showed decreased expression of *HAS3*. Compared against 0% strain expression levels, 10% strain had a significantly higher fold difference of *HAS3* than 20% strain (p = 0.043, effective fold difference vs. 0% strain: -1.235 ± 0.212). Cells strained to 5% strain (p = 0.050, effective fold difference vs. 0% strain: -1.957 ± 0.189) and 20% strain (p = 0.045, effective fold difference vs. 0% strain: -1.822 ± 0.096) strain showed a significantly lower in *HYAL1* expression than the 0% strain group. Similar trends are seen in *TMEM* expression, with 5% strain (p = 0.008, effective fold difference vs. 0% strain: -2.075 ± 0.100) and 20% strain (p = 0.030, effective fold difference vs. 0% strain: -2.370 ± 0.068) groups having lower expression than 0% strain. 10% strain had a higher *TMEM* fold difference than 20% strain (p = 0.024, fold difference vs. 0% strain: -1.433 ± 0.040).

To determine the largest source of variation within gene expression and cytokine concentration data, we used sPCA and identified the Principal Components (PCs) that captured the largest axes of variance within the data. sPCA of gene expression values revealed that PC1 and

PC2 explained 99% of the variability within the data). *HAS1* provided the greatest weight for PC1, while *HAS3*, *TMEM*, *HYAL2*, and *HYAL1* influenced PC2 the greatest (Fig 3B). When clustering by strain group, gene expression data organizes well with distinct clusters for each strain group. The clusters vary along the PC2 axis.

## Hyaluronan quantification and localization

All loading groups had a lower hyaluronan concentration than of the unloaded (0% strain) group (Fig 4A). 10% strain resulted in the highest concentration among loaded groups. Immunofluorescence imaging of the cell layers revealed that all stained hyaluronan was localized intracellularly (Fig 4C), with no staining within the pericellular or extracellular matrices. Quantification of fluorescent staining of hyaluronan showed no significant differences between groups (Fig 4B).

## PRG4 analysis

PRG4 content was measured from media obtained 24 hours after the end of loading. 5% strain resulted in a significantly higher percent difference of PRG4 concentration compared to the 10% strain and 20% strain groups. Interestingly, 5% strain was the only group that resulted in an increase in PRG4 concentration compared to the unloaded controls, while 10% strain and 20% strain reduced the amount of PRG4. 5% strain resulted in a 26.3% ± 0.37 increase in PRG4 concentration compared to 0% strain. 10% strain resulted in a 64.3% ± 0.20 decrease compared to 0% strain and 20% strain resulted in a 38.2% ± 0.10 decrease compared to 0% strain. 0% strain had a higher PRG4 concentration ($p > 0.05$) than the TCP group (S2 Fig).

## Cytokine profile

Synoviocytes cultured on silicone membranes produced significantly higher cytokine concentrations when compared to media from the TCP group (Fig 5A, S3 Fig, S4 Table). The cytokines VEGF, IL-1ra, IL-1B, IL-10, IL-12, IL-13, IL-15, IL-17, and IL-2 were excluded from analysis due to concentrations falling outside of the standard curve. Percent difference in cytokine concentrations between loaded groups and 0% strain show distinct patterns between strain magnitudes (Fig 5B). 5% strain had a significantly higher GM-CSF concentration than 0% strain ($p = 0.007$, +110.56%) and a significantly lower CCL5 concentration than 10% ($p = 0.029$, -62.04%) and 20% strain ($p = 0.017$, -64.27%). 5% strain had a lower eotaxin concentration compared to 10% ($p = 0.010$, -40.32%) and 20% strain ($p = 0.041$, -33.77%) and a lower IP-10 concentration compared to 10% strain ($p = 0.030$, -64.65%) and 20% strain ($p = 0.005$, -72.62%). 10% strain did not significantly change cytokines compared to 0% strain. 20% strain increased GM-CSF ($p = 0.001$, +160.29%) and IP-10 ($p = 0.0383$, +135.28%) concentrations compared to 0% strain (S3 Fig).

sPCA of cytokine concentration also revealed that PC1 and PC2 explained 99% of the variance within the data. PC1 was most heavily influenced by IL-8 while IL-6 contributed to PC2 the most (Fig 5C). Cytokine concentration did not create distinct clusters amongst strain groups, with 0% strain and 20% strain having the most distinction along PC2.

## Multivariate analysis

Characterizing cellular responses of synoviocytes to physical strain is challenging due to the correlation between biomolecules. To deconvolute the contribution of mechanical strain to these complex cellular responses, we employed multivariate data-driven modeling to characterize the relationship between mechanical perturbation and the holistic response of synovial

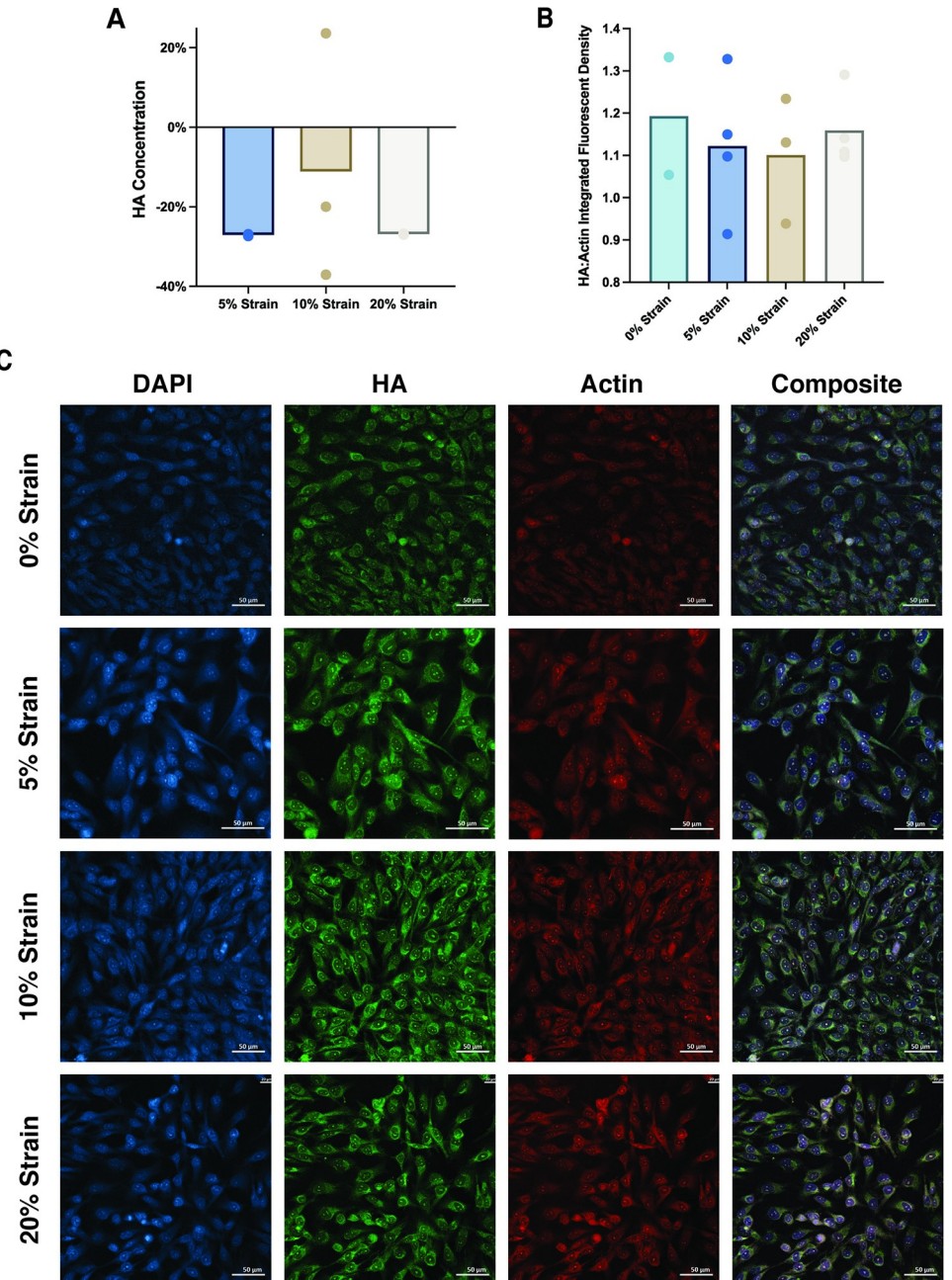

**Fig 4. Hyaluronan content is sensitive to changes to cyclic strain.** A) Overall hyaluronan concentration (ng/mL). B) Integrated density of hyaluronan staining (green, Alexa Fluor-488) normalized to fluorescent intensity of actin staining (red, Phalloidin-565). C) Representative immunofluorescent images of DAPI, hyaluronan, and F-actin stains and a composite image of all strain groups. Dots represent individual samples and significant differences between groups are shown with brackets. Scale bar = 50 μm.

cell gene expression and cytokine production. Cyclic strain impacts hyaluronan gene expression and ultimately hyaluronan concentration, while also affecting cytokine concentration.

We used sparse Partial Least Squares discriminant analysis (sPLSDA) and regression (sPLSR) to explore the relationships between mechanical strain, cytokine production, and gene expression. We trained 6 PLS models in total, predicting mechanical strain (sPLSDA),

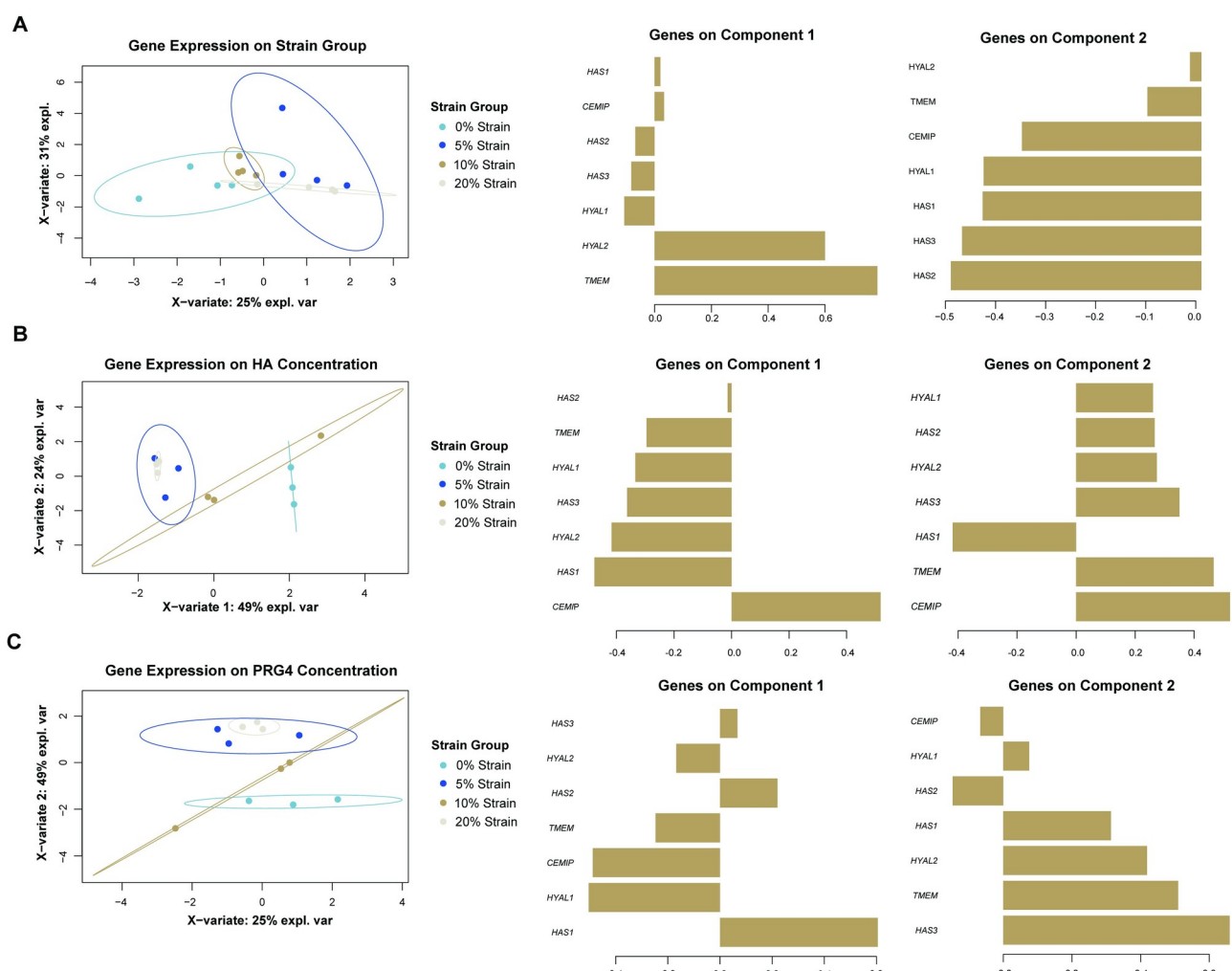

**Fig 5. Substrate stiffness and strain magnitude display distinct cytokine concentration profiles in conditioned media.** A) Values shown are mean percent difference cytokine concentrations of loaded media samples compared to the TCP group. B) Values shown are mean percent difference cytokine concentrations of loaded media samples compared to the 0% strain group (blue indicates a percent decrease, red indicates a percent increase). C) Sparse PCA of cytokine concentration with each sample grouped by strain group, along with corresponding loadings plots. sPCA is shown as a scatter plot in the X-Y representative space of the gene expression data along PC1 and PC2, which explain over 99% of the variation in the data.

hyaluronan concentration and PRG4 concentration (sPLSR) from either cytokine concentration (3 models) or hyaluronan gene expression (3 models) data.

When predicting strain group from gene expression data, we found that component 1 distinguishes between strain groups well (Fig 6A, S4A Fig). Component 1 is most influenced by *TMEM* and *HYAL2*. We found that gene expression separated hyaluronan concentration data between 0% strain and 5% strain along Component (Fig 6B, S5A Fig). Component 1 was most influenced by *CEMIP* and *HAS1*. Gene expression latent variables also separated 0% strain and 5% strain in our model of PRG4 concentration (Fig 6C, S5B Fig). Component 1 was influenced by *HAS1 HYAL1*, and *CEMIP* and component 2 was contributed by *HAS3*, *TMEM*, and *HYAL2*.

In contrast to the gene expression data, we find overall stronger relationships between cytokine production, PRG4 concentration, hyaluronan concentration, and mechanical strain. Clusters when projecting cytokine concentration on strain group were across both component

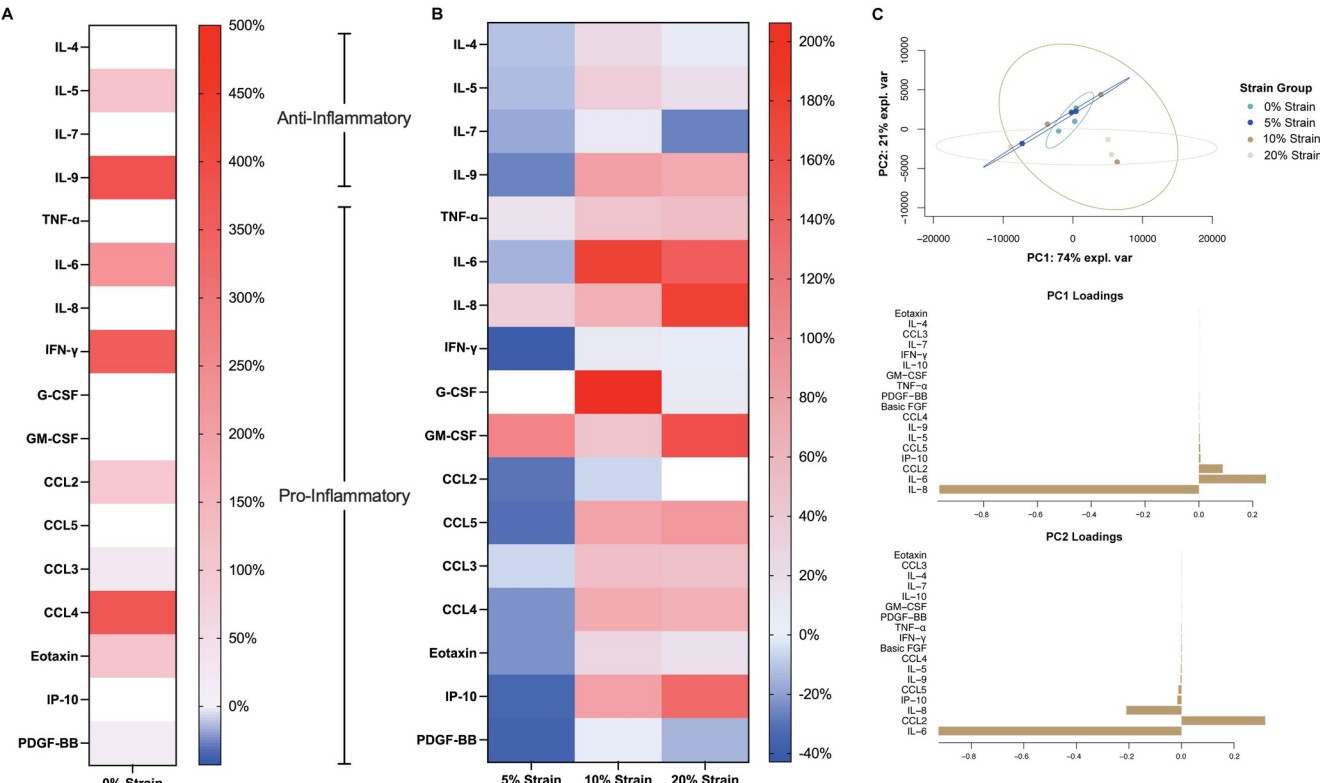

**Fig 6. Sparse partial least squares (sPLS) analysis of hyaluronan-related gene expression to predict strain group, hyaluronan concentration, and PRG4 concentration show distinct clusters between 0% and 5% strain.** A) sPLS-DA to predict strain group from gene expression data, along with corresponding loadings plots, showing the contributions of each gene for each component. B) sPLSR to predict hyaluronan concentration group from gene expression data, along with corresponding loadings plots, showing the contributions of each gene for each component. C) sPLSR to predict PRG4 concentration from gene expression data, along with corresponding loadings plots, showing the contributions of each gene for each component. sPLS plots are shown in the X representative space of data along 2 components.

1 and 2. Component 2 separated out 5% strain, while component 1 separated out 0% strain and 10% strain well (Fig 7A, S4B Fig). Component 1 was influenced by eotaxin, CCL5, and IL-9, while component 2 was influenced by GM-CSF, IL-10, and CCL2. When projecting cytokine concentration onto hyaluronan concentration, there are distinct clusters amongst the 0% strain and 5% strain groups along component 1 (Fig 7B, S5C Fig). Component 1 was most heavily influenced by GM-CSF and CCL2. When projecting cytokine concentration on PRG4 concentration, component 1 separated 5% strain and 10% strain and component 2 separated 5% and 20% strain (Fig 7C, S5D Fig). CCL4 and eotaxin weighed heavily for component 1, while FGF and CCL2 influenced component 2 the most.

## Discussion

Our goal was to determine how varying cyclic strain magnitudes will affect synoviocyte production of hyaluronan, PRG4, and cytokines. We performed these experiments in SW982 cells, a synoviocyte cell line that has been used as *in vitro* inflammatory cell models for OA studies [52, 53], and evaluated how sensitive hyaluronan and PRG4 production is to both changes in substrate and different levels of cyclic tensile strains in synovial fibroblasts.

Our results did not show a linear magnitude-dependent relationship with strain. We expected that higher strain magnitudes would increase production of hyaluronan and PRG4 and decrease production of pro-inflammatory cytokines; however, we rejected this hypothesis

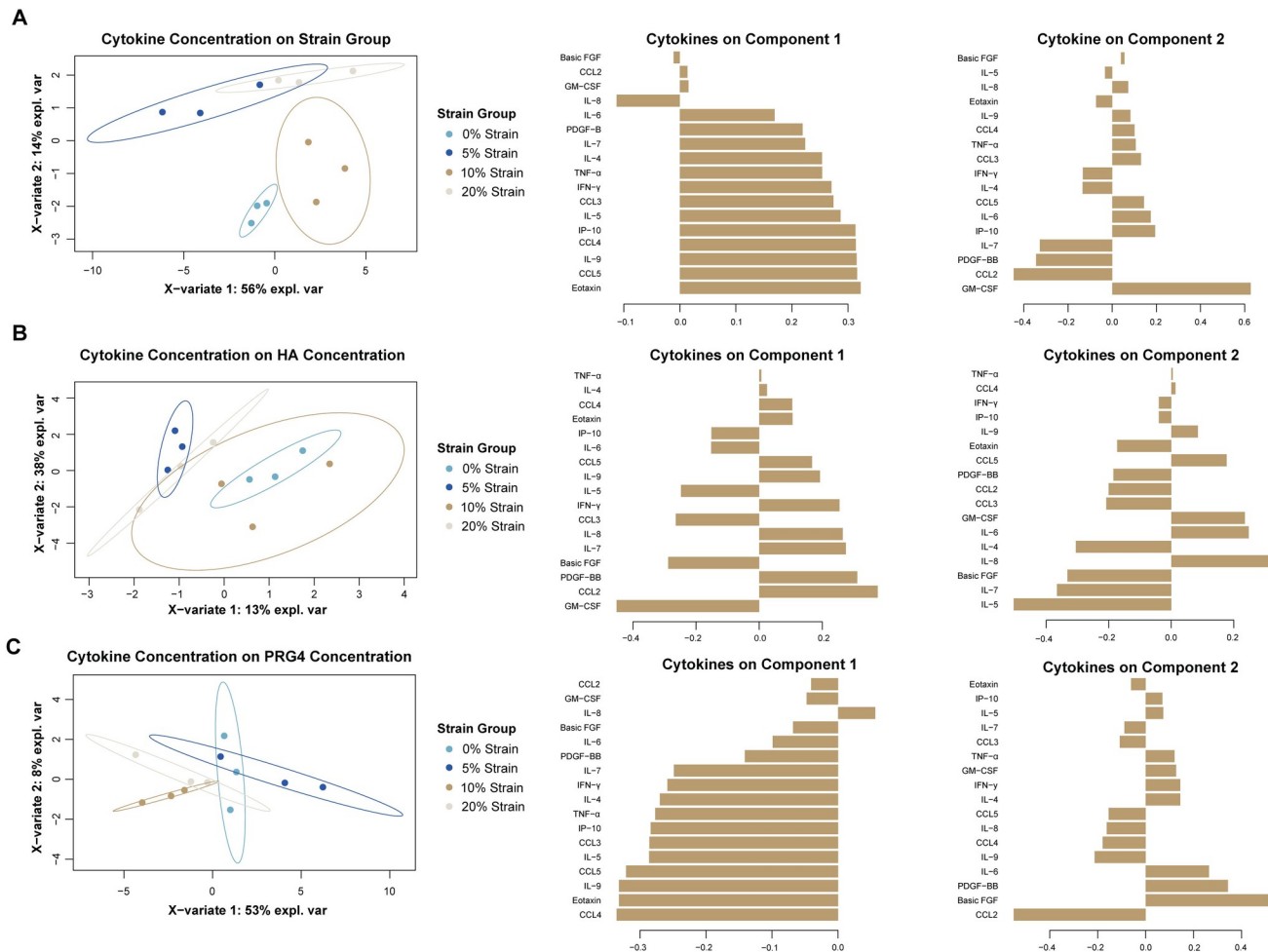

**Fig 7. sPLS analysis of cytokine concentration to predict strain group, hyaluronan concentration, and PRG4 concentration shows distinct clustering between strain groups.** A) sPLS-DA to predict strain group from cytokine concentration data, along with corresponding loadings plots, showing the contributions of each cytokine for each component. B) sPLSR to predict hyaluronan concentration group from cytokine concentration data, along with corresponding loadings plots, showing the contributions of each cytokine for each component. C) sPLSR to predict PRG4 concentration from cytokine concentration data, along with corresponding loadings plots, showing the contributions of each cytokine for each component. sPLS plots are shown in the X representative space of data along 2 components.

based on the results. While hyaluronan synthase and hyaluronidase gene expression in synoviocytes were sensitive to cyclic strain, overall hyaluronan concentration present in the media did not reflect the change in expression–cyclic stretch reduced overall hyaluronan content within the conditioned media. Low, but not high, strain magnitudes increase PRG4 concentration and have a more anti-inflammatory cytokine profile in the media compared to an unloaded control.

We saw no significant changes to HAS gene expression with a change in substrate stiffness. Others have found that static stretch of synovial fibroblasts resulted in minimal change to *HAS2* and *HAS3* expression [40]. Static stretch effectively increases the stiffness of a substrate, our observations with substrate stiffness are consistent with this prior work in static stretch. Although the effect of changes to substrate stiffness on hyaluronan production has not been thoroughly studied, especially with synovial fibroblasts, chondrocytes have been shown to produce more aggrecan, collagen, and proteoglycans when seeded onto softer substrates [54, 55].

While the addition of mechanical stimulation showed significant differences in *HAS3* expression, there were no significant differences among strain magnitudes. In this study, 5% and 20% strain of synoviocytes resulted in lower *HAS3* expression, unlike static stretch, indicating that synoviocytes may be more responsive to cyclic stimulation as opposed to static stretch or a change in substrate stiffness. Our results on synoviocytes under cyclic loading are supported by prior work showing that cyclic compression of synoviocytes upregulates expression of *HAS2* and *HAS3* [31]. Together, this study in tension and others in compression suggest that hyaluronan turnover by synoviocytes is responsive to cyclic mechanical stimulation.

Our data suggest that 5% and 20% strain increase the production of lower molecular weight hyaluronan, due to an increase in *HAS3* expression, and increase hyaluronan breakdown, due to an increase in hyaluronidase expression, indicating that the molecular weight distribution may be shifted downward. *HYAL1*, *HYAL2*, and *TMEM* expression was upregulated in a strain magnitude-dependent manner. This is the first report of the strain-sensitivity of hyaluronidase expression. Although there was no significant difference among media from loaded groups, overall hyaluronan was less than that of the unloaded control. This is supported by the localized intracellular hyaluronan found in immunofluorescence imaging. Synthesized hyaluronan from synoviocytes are released into the synovial fluid [56], which supports our findings of the absence of pericellular or extracellular hyaluronan staining found in our images. Intracellular hyaluronan is indicative of hyaluronan reuptake and breakdown [57] and as a marker for endoplasmic reticulum stress [58], coinciding with the upregulation of hyaluronidase expression in our loaded groups. Future studies that inhibit hyaluronan synthases or hyaluronidases to determine what pathway is most influential on final hyaluronan concentration would give more insight on mechanotransduction of hyaluronan in synoviocytes.

Additionally, a low strain magnitude improved production of PRG4, while high strain magnitudes decreased PRG4 production. PRG4 secretion has been shown to increase in the presence of dynamic shear of cartilage tissue [59] and in cyclic compression [60, 61]. Cyclic tensile strain has also been shown to upregulate PRG4 expression in chondrocytes. Reduction of high molecular weight hyaluronan inhibits the boundary lubrication ability of synovial fluid, but addition of PRG4 to a hyaluronan solution can reduce friction [62]. This suggests that low strain mechanical stimulation of synovial fibroblasts could also be a good target to improve PRG4 synthesis to alleviate changes to synovial fluid lubrication.

We used multivariate analysis to better understand the relationship between gene expression and hyaluronan concentration, PRG4 concentration, and strain group. Through sPCA analysis, we saw strain group clusters along PC2, which is heavily influenced by *HAS1*. This implies that *HAS1* expression may lead to distinct expression clusters between strain groups, which is interesting because univariate statistical testing did not identify significant differences between groups with *HAS1* expression. Although univariate statistical testing also did not identify any significant differences between hyaluronan concentrations, sPLSR analysis suggested that *HAS1* and *CEMIP* expression contributed the most to distinct changes in hyaluronan concentrations between 0% and 5% strain. sPLSR analysis also showed that *HAS3* and *TMEM* may have influenced the separation between 0% and 5% strain. Overall, sPLS analysis show clear distinction between 0% and 5% strain group when using gene expression to predict other variables in this study, highlighting the addition of cyclic strain, specifically low magnitude, affects hyaluronan and PRG4 concentration.

We show that 5% strain seems to decrease the concentration of cytokines and chemokines that are typically elevated in pathological joints, while higher strain magnitudes seem to increase pro-inflammatory cytokines.5% strain reduced concentrations of pro-inflammatory markers such as IFN-γ, IL-6, and TNF-α, while 10% and 20% strain increased concentrations of these cytokines. IL-6, IL-8, TNF-α levels have been shown to be elevated in synovial fluid of

patients with OA and RA and in post-injury joints, suggesting that high strain magnitudes are pushing towards a more po-inflammatory environment [63–65]. IL-9, which has been shown to stimulate pathologic T-cells within the joint [66], was also elevated in the media from cells stretched to 10% and 20% strain. Macrophage inflammatory protein-1 beta (MIP-1β or CCL4) [67] and CCL5 [68] have also been shown to be elevated in synovial fluid samples of patients with OA. Here, we see reduced levels of CCL4 and CCL5 in the media from the 5% strain group. Previous studies have seen that the effect of cyclic strain on cytokine production, specifically in synovial cells, is dependent on the loading parameters. Short-term exposure (4h) to loads result in an increase in IL-6 and TNF-α [38], while long-term exposure (24h) at a slower strain rate also resulted in elevated IL-8 levels [69]. The variability in cytokine production from different loading parameters highlights the sensitivity of synoviocytes to different loading parameters.

We probed interdependency among hyaluronan concentration, PRG4 concentration, and cytokine concentration using multivariate analysis. sPCA did not show distinct clusters or patterns among cytokine concentration data when grouping by hyaluronan concentration, PRG4 concentration, or strain group. sPLS did show better organization when projecting cytokine concentration on to strain group, hyaluronan concentration and PRG4 concentration. Since sPCA is a method that is descriptive of the data, this indicates that while cytokine concentration data does not distinguish between groups itself, it can be used inferentially to project data. Both strain group and the resulting hyaluronan concentration separated 0% and 5% strain in cytokine concentration data. Cytokine concentration data organized PRG4 concentration data well, with CCL4 and eotaxin separating out 5% and 10% strain and CCL2 and FGF separating 10% and 20% strain. sPLS analysis highlights our previous finding that stain magnitude results in distinct cytokine profiles and more work needs to be done to elucidate the mechanism between synovial mechanical stimulation and cytokine production.

Our study has several limitations. As noted above, we used a synoviocyte cell line. Primary cells may behave differently, so there should be future efforts to evaluate the relationship between cyclic loading and the production of synovial fluid components from both healthy and diseased tissues. Second, we did not measure hyaluronan molecular weight distribution in our study. Despite the changes in hyaluronan-related gene expression, we did not see changes in overall hyaluronan concentrations–suggesting that hyaluronan molecular weight could be affected. Future studies would benefit from assessing how cyclic tensile strain impacts hyaluronan molecular weight. Finally, we subjected synoviocytes to short-term exposure of cyclic strain. Thus, it is possible that our model is conservative and more substantial effects for hyaluronan turnover and concentration would occur at longer exposure times.

In conclusion, we show that cyclic loading can be used to affect hyaluronan turnover, PRG4, and cytokine concentration. Changes in substrate stiffness result in significant changes in cytokine production from synoviocytes. While an increase in strain magnitude decreased PRG4 concentration and altered production of cytokines, hyaluronan concentration was unchanged by differences in magnitude. Our results show that mechanical activity of the synovium could be a key regulator of the joint environment and understanding the relationship between mechanics and joint health can help identify new therapeutic targets for pathology.

## Supporting information

**S1 Fig. Relative transcript abundance was calculated from ΔCq values, with *GAPDH* as a housekeeping gene.** Black circles represent individual samples with significant differences between groups are shown with brackets.
(PDF)

**S2 Fig. Concentrations of PRG4 concentrations for all groups.** Black circles represent individual samples with significant differences between groups are shown with brackets. (PDF)

**S3 Fig. Cytokine concentrations for all groups.** Black circles represent individual samples with significant differences between groups are shown with brackets. Significance between the TCP and 0% strain group was established using a t test or non-parametric alternative. Significance between the 0%, 5%, 10%, and 20% strain groups was established using a one-way ANOVA or non-parametric alternative. Concentrations are shown as pg/mL. (PDF)

**S4 Fig. Performance statistics for sPLS-DA analyses.** Classification error rates and ROC curves of each component for A) gene expression and B) cytokine concentration predicting strain group. (PDF)

**S5 Fig. Performance statistics for sPLSR analyses.** $Q^2_{total}$ values of each component for A) gene expression predicting hyaluronan concentration, B) gene expression predicting PRG4 concentration, C) cytokine concentration predicting hyaluronan concentration, and D) cytokine concentration prediction PRG4 concentration. (PDF)

**S1 Table. Inventoried TaqMan® (Invitrogen) probes used for qRT-PCR.** (PDF)

**S2 Table. Average Cq values for GADPH of each experimental groups and fold difference values against the 0% strain group.** (PDF)

**S3 Table. Average Cq values ± standard deviation for each gene of interest of each experimental group.** (PDF)

**S4 Table. Raw cytokine concentration values for each sample.** Zeroes indicate a value that fell below the lower limit of the standard curve. All cytokines are shown as pg/mL. (PDF)

## Acknowledgments

The authors acknowledge Maya Federle, Megan Baker, and Javier Munoz for acquiring cytokine concentration data and Mary Cowman for guidance on hyaluronan ELISA sample prep. The authors would also like to thank Clarisse Zigan, Neeharicka Measala, and Aritra Chatterjee for confocal image acquisition. The authors also acknowledge Lubris BioPharma for providing the rhPRG4 in kind. This material is based upon work supported by the National Science Foundation under Grant No. 1944394 and 2149946.

## Author Contributions

**Conceptualization:** Meghana Pendyala, Douglas K. Brubaker, Deva D. Chan.

**Data curation:** Meghana Pendyala, Paige S. Woods.

**Formal analysis:** Meghana Pendyala.

**Funding acquisition:** Deva D. Chan.

**Investigation:** Meghana Pendyala, Paige S. Woods.

**Project administration:** Douglas K. Brubaker, Tannin A. Schmidt, Deva D. Chan.

**Resources:** Douglas K. Brubaker, Elizabeth A. Blaber, Tannin A. Schmidt, Deva D. Chan.

**Software:** Meghana Pendyala, Douglas K. Brubaker.

**Supervision:** Douglas K. Brubaker, Elizabeth A. Blaber, Deva D. Chan.

**Validation:** Meghana Pendyala, Douglas K. Brubaker, Elizabeth A. Blaber, Tannin A. Schmidt, Deva D. Chan.

**Visualization:** Meghana Pendyala.

**Writing – original draft:** Meghana Pendyala.

**Writing – review & editing:** Meghana Pendyala, Paige S. Woods, Douglas K. Brubaker, Elizabeth A. Blaber, Tannin A. Schmidt, Deva D. Chan.

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
