## [Decision Letter · Decision Letter 0]

14 Aug 2022

PONE-D-22-10694Endogenous production of hyaluronan, PRG4, and cytokines is sensitive to cyclic loading in synoviocytesPLOS ONE

Dear Dr. Chan,

Thank you for submitting your manuscript to PLOS ONE. After careful consideration, we feel that it has merit but does not fully meet PLOS ONE’s publication criteria as it currently stands. Therefore, we invite you to submit a revised version of the manuscript that addresses the points raised during the review process.

We look forward to receiving your revised manuscript.

Kind regards,

Jianhong Zhou

Staff Editor

PLOS ONE

Journal Requirements:

Reviewers' comments:

Reviewer's Responses to Questions

**Comments to the Author**

1. Is the manuscript technically sound, and do the data support the conclusions?

Reviewer #1: Yes

2. Has the statistical analysis been performed appropriately and rigorously? 

Reviewer #1: Yes

3. Have the authors made all data underlying the findings in their manuscript fully available?

Reviewer #1: Yes

4. Is the manuscript presented in an intelligible fashion and written in standard English?

Reviewer #1: Yes

5. Review Comments to the Author

Reviewer #1: The work by Pendyala and collaborators investigates whether cyclic loading will alter hyaluronan and PRC4 content, hyaluronan metabolizing enzymes, and cytokines expression. Interestingly the authors found a decreased hyaluronan content and alteration of pro and anti-inflammatory cytokines.

Although the manuscript is interesting, the authors should answer some concerns.

1. The gene expression was performed after 4 hours from the end of loading. Why do the authors choose this time point? Have the authors tried other time points as 8 or 16 hours after the end of loading? I suspect that 4 hours is not enough to detect a variation in gene expression.

2. Results, first paragraph. I suggest to the authors add a couple of sentences to explain the experiment. Experts in the field should probably skip the introduction and methods and would be able to catch the point.

3. Hyaluronan staining. Have the authors tried to visualize the pericellular matrix by particle exclusion assay? In fig 4 it is not clear the amount of hyaluronan in the pericellular space as it seems intracellular.

4. Hyaluronan quantification. I suggest adding to figure 4A the quantification shown in supplemental fig. S2 clearly reports the concentration of hyaluronan in samples.

5. In the last sentence of the Introduction the authors hypothesized the results they expected. In the discussion, the authors should report whether or not their hypothesis was true or false.

6. It is not clear whether the reduction of hyaluronan after the cyclic loading is due to a reduction of HASes or induction of HYAL or TMEM. Have the authors tried to inhibit HASes or TMEM to identify the mechanism?

6. PLOS authors have the option to publish the peer review history of their article (what does this mean?). If published, this will include your full peer review and any attached files.

Reviewer #1: No

---

## [Author Response · Author response to Decision Letter 0]

20 Sep 2022

These responses are also included in the cover letter as requested.

1. The gene expression was performed after 4 hours from the end of loading. Why do the authors choose this time point? Have the authors tried other time points as 8 or 16 hours after the end of loading? I suspect that 4 hours is not enough to detect a variation in gene expression.

R1.1. Previous studies that have measured changes to expression of hyaluronan synthases after mechanical stimulation have done so at varying time points. Momberger et. al., measured hyaluronan synthase expression from synovial fibroblasts 3 hours after the end of loading (3h of static stretch). They found that Has2 expression increases with static stretch, even at just 3h after the end of loading. 

Studies have shown that changes to hyaluronan synthase expression can occur as early as immediately after stimulation in different cell types, supporting our choice to collect total RNA 4h after the end of loading. Chondrocytes stimulated with TGF-β1 showed immediate upregulation of Has2 following simulation. In lutein cells, mRNA collected immediately after varying TGF- β1 stimulation times resulted in significant upregulation of HAS2 and HAS3 {Wang, 2019, 31437481}. 

We have included a statement in the methods to make this clear to the reader.

"To evaluate the early gene expression change in response to loading, media was removed from 3 wells and cell layers were lysed and pooled for RNA isolation, as described below, four hours after the end of loading. Prior work has demonstrated that changes to hyaluronan synthase expression can occur as early as immediately after external stimulation [40, 41]."

2. Results, first paragraph. I suggest to the authors add a couple of sentences to explain the experiment. Experts in the field should probably skip the introduction and methods and would be able to catch the point.

R1.2. As suggested, a paragraph has been included at the beginning of the Results section to quickly summarize the experiment. 

"To assess the synoviocyte response to a change in substrate stiffness and 5%, 10%, and 20% cyclic strain at 1 Hz, we used qPCR to measure changes in expression of hyaluronan synthesis and breakdown genes, ELISAs to measure hyaluronan and PRG4 concentrations, and a multiplex panel to measure cytokine concentrations in media."

3. Hyaluronan staining. Have the authors tried to visualize the pericellular matrix by particle exclusion assay? In fig 4 it is not clear the amount of hyaluronan in the pericellular space as it seems intracellular.

R.1.3. The reviewer is correct in that the hyaluronan we observed was indeed only found intracellularly and not as pericellular matrix. To help clarify this point, additional text has been included in the Results section to specify that immunofluorescent staining of hyaluronan was localized only intracellularly, not in the pericellular or extracellular matrices.

"Immunofluorescence imaging of the cell layers revealed that all stained hyaluronan was localized intracellularly (Fig 4C), with no staining within the pericellular or extracellular matrices."

In addition, further discussion is included to explain why we may have found only intracellular staining.

"This is supported by the localized intracellular hyaluronan found in immunofluorescence imaging. Synthesized hyaluronan from synoviocytes are released into the synovial fluid [56], which supports our findings of the absence of pericellular or extracellular hyaluronan staining found in our images. Intracellular hyaluronan is indicative of hyaluronan reuptake and breakdown [57] and as a marker for endoplasmic reticulum stress [58], coinciding with the upregulation of hyaluronidase expression in our loaded groups."

Since we expected synovial fibroblast cells to secrete hyaluronan into the media, we were not expecting hyaluronan in the pericellular coat – which is why we did not perform a particle exclusion assay. Particle exclusion assays for hyaluronan must be done in sparse cell cultures to visualize the pericellular coat . Our study was done at ~80% confluency in the monolayer in order to measure our other endpoints. 

4. Hyaluronan quantification. I suggest adding to figure 4A the quantification shown in supplemental fig. S2 clearly reports the concentration of hyaluronan in samples.

R1.4. Figure 4A and Figure S2 have been adjusted as suggested. 

5. In the last sentence of the Introduction the authors hypothesized the results they expected. In the discussion, the authors should report whether or not their hypothesis was true or false.

R1.5. A clear statement on whether our hypothesis was true or false has been included.

Our results did not show a linear magnitude-dependent relationship with strain. We expected that higher strain magnitudes would increase production of hyaluronan and PRG4 and decrease production of pro-inflammatory cytokines; however, we rejected this hypothesis based on the results.

6. It is not clear whether the reduction of hyaluronan after the cyclic loading is due to a reduction of HASes or induction of HYAL or TMEM. Have the authors tried to inhibit HASes or TMEM to identify the mechanism?

R1.6. While we did not try to inhibit any of the synthases of hyaluronidases, the authors acknowledge this would be insightful to determine the mechanism behind the effect of cyclic loading on hyaluronan concentration. Small interfering RNA has been used to inhibit HAS2, HAS3, and HYALs and typical protocols include incubating the siRNA for over 24 hours to ensure knockdown of the gene of interest [Lai, 2010, PMID: 20305449; Li, 2007, PMID: 17315194]. However, this method of inhibition has not been done in synovial fibroblasts. 

Inhibiting either HASes or HYALs would require a set of pilot studies to determine delivery mechanism and the incubation time necessary to knockdown these genes [Haiyong, 2018, PMID: 29423805]. Pilot studies would also have to assess whether the transfection process affects the mechanorepsonsiveness of the cells, since electroporation may be necessary in this cell line. The scope of the current study was to determine how cyclic strain magnitude affects hyaluronan production and the balance of synthase and hyaluronidase expression. Having now established that the balance of synthesis and breakdown are not linear with respect to strain magnitude, future studies can then be designed to evaluate the mechanisms underlying these changes and associated mechanotransduction pathways. We have included this discussion in the text:Future studies that inhibit hyaluronan synthases or hyaluronidases to determine what pathway is most influential on final hyaluronan concentration would give more insight on mechanotransduction of hyaluronan in synoviocytes.

---

## [Decision Letter · Decision Letter 1]

13 Oct 2022

PONE-D-22-10694R1Endogenous production of hyaluronan, PRG4, and cytokines is sensitive to cyclic loading in synoviocytesPLOS ONE

Dear Dr. Chan,

Thank you for submitting your manuscript to PLOS ONE. After careful consideration, we feel that it has merit but does not fully meet PLOS ONE’s publication criteria as it currently stands. Therefore, we invite you to submit a revised version of the manuscript that addresses the final minor points that were raised by Reviewer#2. 

We look forward to receiving your revised manuscript.

Kind regards,

Andre van Wijnen

Academic Editor

PLOS ONE

Journal Requirements:

Additional Editor Comments:

Pendyala et al. studied various strains of cyclic loading on synoviocytes on hyaluronan, PRG4, and cytokines expression. SW982 (synovial sarcoma) synoviocytes were cyclically loaded to 0%, 5%, 10%, or 20% strain for three hours to assess the impact of substrate stiffness, compared with the 0% strain group. Expression of hyaluronan turnover genes, hyaluronan localization within the cell layer, hyaluronan concentration, PRG4 concentration, and the cytokine profile within the media. The authors conclude that the cyclic loading increased HAS3 expression (showed magnitude-independent response). However, strain magnitude impacted hyaluronidase expression and was associated with decreased hyaluronan concentration. Finally, the 10% and 20% strains show a distinct, pro-inflammatory cytokine profile compared to the unloaded group.

Major points:

1) Many of the cytokines Il1Ra and IL-1b are reported to be responsive to cyclic strains. However, these cytokines were out of the standard curve in the current study. Are these cytokines falling above or below standard curve? If it is above, why was dilution not considered for estimating the concentration?

2) mRNA expression of these cytokines could also have been studied.

3) Since synoviocytes cell line (tumor cells) was used – cyclic strains are density dependent - how was cell density optimized for the study?

4) Please provide the table with the concentration of various cytokines (minimum, maximum). The cytokine is expressed ng or pg/ul rather than normalized to protein concentration. Did 10% or 20% cyclic strains induce cell death?

Reviewers' comments:

Reviewer's Responses to Questions

**Comments to the Author**

1. If the authors have adequately addressed your comments raised in a previous round of review and you feel that this manuscript is now acceptable for publication, you may indicate that here to bypass the “Comments to the Author” section, enter your conflict of interest statement in the “Confidential to Editor” section, and submit your "Accept" recommendation.

Reviewer #1: All comments have been addressed

Reviewer #2: (No Response)

2. Is the manuscript technically sound, and do the data support the conclusions?

Reviewer #1: Yes

Reviewer #2: Yes

3. Has the statistical analysis been performed appropriately and rigorously? 

Reviewer #1: Yes

Reviewer #2: I Don't Know

4. Have the authors made all data underlying the findings in their manuscript fully available?

Reviewer #1: Yes

Reviewer #2: No

5. Is the manuscript presented in an intelligible fashion and written in standard English?

Reviewer #1: Yes

Reviewer #2: Yes

6. Review Comments to the Author

Reviewer #1: All the concerns raised during the first round of revision have been properly addressed by the authors.

Reviewer #2: Pendyala et al. studied various strains of cyclic loading on synoviocytes on hyaluronan, PRG4, and cytokines expression. SW982 (synovial sarcoma) synoviocytes were cyclically loaded to 0%, 5%, 10%, or 20% strain for three hours to assess the impact of substrate stiffness, compared with the 0% strain group. Expression of hyaluronan turnover genes, hyaluronan localization within the cell layer, hyaluronan concentration, PRG4 concentration, and the cytokine profile within the media. The authors conclude that the cyclic loading increased HAS3 expression (showed magnitude-independent response). However, strain magnitude impacted hyaluronidase expression and was associated with decreased hyaluronan concentration. Finally, the 10% and 20% strains show a distinct, pro-inflammatory cytokine profile compared to the unloaded group.

Major points:

1) Many of the cytokines Il1Ra and IL-1b are reported to be responsive to cyclic strains. However, these cytokines were out of the standard curve in the current study. Are these cytokines falling above or below standard curve? If it is above, why was dilution not considered for estimating the concentration?

2) mRNA expression of these cytokines could also have been studied.

3) Since synoviocytes cell line (tumor cells) was used – cyclic strains are density dependent - how was cell density optimized for the study?

4) Please provide the table with the concentration of various cytokines (minimum, maximum). The cytokine is expressed ng or pg/ul rather than normalized to protein concentration. Did 10% or 20% cyclic strains induce cell death?

7. PLOS authors have the option to publish the peer review history of their article (what does this mean?). If published, this will include your full peer review and any attached files.

Reviewer #1: No

Reviewer #2: No

---

## [Author Response · Author response to Decision Letter 1]

18 Nov 2022

The following responses are copied from the cover letter that includes our full response:

Each point has been addressed and changes to the original manuscript are in a Track Changes version. To easily identify changes in the Track Changes document, edited portions of the text are annotated with a red-outlined box, labeled by comment number (e.g., R2.1).

Comments:

1. Many of the cytokines Il1Ra and IL-1b are reported to be responsive to cyclic strains. However, these cytokines were out of the standard curve in the current study. Are these cytokines falling above or below standard curve? If it is above, why was dilution not considered for estimating the concentration?

R2.1. Cytokines that were not included in the analysis, including IL1-Ra and IL-1b, fell below the standard curve. This detail has been updated in the text: 

Cytokines whose concentrations were out of range fell below the lower limit for all samples and were excluded from analyses.

2. mRNA expression of these cytokines could also have been studied.

R2.2. Indeed, gene expression of these cytokines could have been measured; however, this study was focused on the availability of these cytokines in the media, enabling inference of cytokines that may be made available in the joint space. Gene expression related to a particular cytokine would have been directly linked to its protein expression, which is the end product that is made available into the media. On the other hand, we opted to measure gene expression of hyaluronan-related genes because multiple gene products contribute to and affect hyaluronan content in the media, with multiple degrees of separation between gene expression and product availability. In the case of hyaluronan availability, while differential expression of Has1-3 genes may be linked to their respective protein (HAS1-3) expression levels, the activity level of these synthase proteins is what directly adds to available hyaluronan.

3. Since synoviocytes cell line (tumor cells) was used – cyclic strains are density dependent - how was cell density optimized for the study?

R2.3. Cell density was optimized to balance adhesion of cells to the silicone membrane with total cells stimulated in each well. Through pilot experiments, we observed that high confluency caused detachment of cell monolayers from the silicone membrane. To maintain attachment, confluency was therefore limited to 70-80%. Specifically, in pilot studies, we varied seeding density between 2x104 and 5x104 cells/mL and found that concentrations above 4x104 cell/mL resulted in detached of the cell layer. 2x104 cells/mL was not enough to isolate enough total RNA without surpassing the number of wells that could be considered technical replicates and therefore pooled for analysis. Therefore, our final seeding density of 3.5x105 cells/mL (3-4x104 cells/cm2) was used. Each well was plated after confirmation of cell numbers using a cell counter. We have added the following to the text to clarify the results of the pilot studies in guiding these ecperiments:

SW982 cells were seeded on silicone membranes (BioFlex plates, Flexcell International Corp, Burlington, NC) at a density of 3-4 × 104 cells/cm2 (optimized during pilot experiments to minimize cell detachment while preserving adequate total RNA yield) and were allowed to attach for 24 hours in 1 mL complete medium.

4. Please provide the table with the concentration of various cytokines (minimum, maximum). The cytokine is expressed ng or pg/ul rather than normalized to protein concentration. Did 10% or 20% cyclic strains induce cell death?

R2.4. We measured protein concentration in each sample and found no significant differences between groups. A table with raw cytokine concentrations is included in Supplementary Table 3 for the interested reader, and complete data of both protein and cytokine concentration are available in our data repository. The supplementary figure shows units as pg/mL (which is the concentration of all standards for each cytokine).

While we did not directly quantify cell death, qualitative assessments throughout the study did not indicate cell death. Notably, previous studies that used cyclically loaded synovial fibroblasts to 10% strain have shown no changes to viability {Bader, 2010, Cytokine}.

---

## [Decision Letter · Decision Letter 2]

6 Dec 2022

PONE-D-22-10694R2Endogenous production of hyaluronan, PRG4, and cytokines is sensitive to cyclic loading in synoviocytesPLOS ONE

Dear Dr. Chan,

Thank you for submitting your manuscript to PLOS ONE. After careful consideration, Reviewer#1 was satisfied by Reviewer#2 had residual minor comments that should be addressed before it is acceptable for publication in PLOS ONE. We invite you to submit a revised version of the manuscript once you have addresses these final points. Please submit your revised manuscript by Jan 20 2023 11:59PM. If you will need more time than this to complete your revisions, please reply to this message or contact the journal office at plosone@plos.org. Please include the following items when submitting your revised manuscript:A rebuttal letter that responds to each point raised by the academic editor and reviewer(s). You should upload this letter as a separate file labeled 'Response to Reviewers'.A marked-up copy of your manuscript that highlights changes made to the original version. You should upload this as a separate file labeled 'Revised Manuscript with Track Changes'.An unmarked version of your revised paper without tracked changes. You should upload this as a separate file labeled 'Manuscript'.If applicable, we recommend that you deposit your laboratory protocols in protocols.io to enhance the reproducibility of your results. Protocols.io assigns your protocol its own identifier (DOI) so that it can be cited independently in the future. For instructions see: https://journals.plos.org/plosone/s/submission-guidelines#loc-laboratory-protocols. Additionally, PLOS ONE offers an option for publishing peer-reviewed Lab Protocol articles, which describe protocols hosted on protocols.io. Read more information on sharing protocols at https://plos.org/protocols?utm_medium=editorial-email&utm_source=authorletters&utm_campaign=protocols.

We look forward to receiving your revised manuscript.

Kind regards,

Andre van Wijnen

Academic Editor

PLOS ONE

Journal Requirements:

Additional Editor Comments (if provided):

REviewer#2:

1) authors have addressed all the comments satisfactorily. However, the concentration of cytokine presented in Supplemental Figure 3 and Supplemental table 3 d o not match. In figure all the cytokine presented as ng/ml. The legends needs to be more informative and include all details.

2) in the supplemental table - authors should provide the average CT for the targets studied.

Reviewers' comments:

Reviewer's Responses to Questions

**Comments to the Author**

1. If the authors have adequately addressed your comments raised in a previous round of review and you feel that this manuscript is now acceptable for publication, you may indicate that here to bypass the “Comments to the Author” section, enter your conflict of interest statement in the “Confidential to Editor” section, and submit your "Accept" recommendation.

Reviewer #1: All comments have been addressed

Reviewer #2: All comments have been addressed

2. Is the manuscript technically sound, and do the data support the conclusions?

Reviewer #1: (No Response)

Reviewer #2: Yes

3. Has the statistical analysis been performed appropriately and rigorously? 

Reviewer #1: (No Response)

Reviewer #2: Yes

4. Have the authors made all data underlying the findings in their manuscript fully available?

Reviewer #1: (No Response)

Reviewer #2: No

5. Is the manuscript presented in an intelligible fashion and written in standard English?

Reviewer #1: (No Response)

Reviewer #2: Yes

6. Review Comments to the Author

Reviewer #1: All concerns raised during the past reviewing processes have been properly addressed by the authors.

Reviewer #2: 1) authors have addressed all the comments satisfactorily. However, the concentration of cytokine presented in Supplemental Figure 3 and Supplemental table 3 d o not match. In figure all the cytokine presented as ng/ml. The legends needs to be more informative and include all details.

2) in the supplemental table - authors should provide the average CT for the targets studied.

7. PLOS authors have the option to publish the peer review history of their article (what does this mean?). If published, this will include your full peer review and any attached files.

Reviewer #1: No

Reviewer #2: No

---

## [Author Response · Author response to Decision Letter 2]

8 Dec 2022

Thank you for the additional comments on our manuscript titled “Endogenous Production of Hyaluronan, PRG4, and Cytokines is Sensitive to Cyclic Loading in Synoviocytes.” Each point has been addressed and changes to the original manuscript are in a Track Changes version. To easily identify changes in the Track Changes document, edited portions of the text are annotated with a red-outlined box, labeled by comment number (e.g., R3.1). Our responses to each reviewer comment are below.

Comments:

R3.1 Authors have addressed all the comments satisfactorily. However, the concentration of cytokine presented in Supplemental Figure 3 and Supplemental table 3 do not match. In figure all the cytokine presented as ng/ml. The legends needs to be more informative and include all details.

Both S3 Figure and S4 Table present the cytokine concentrations in pg/mL. This detail has been made explicit in the caption for S3 Figure to be more informative. We have also checked that all supplemental information is clearly referred to within the main body of the text.

R3.2. In the supplemental table - authors should provide the average CT for the targets studied.

We have included an additional supplementary table (Table S3) that includes the average Cq value for each gene of interest for every group.

---

## [Editor Report · Decision Letter 3]

12 Dec 2022

Endogenous production of hyaluronan, PRG4, and cytokines is sensitive to cyclic loading in synoviocytes

PONE-D-22-10694R3

Dear Dr. Chan,

We’re pleased to inform you that your manuscript has been judged scientifically suitable for publication and will be formally accepted for publication once it meets all outstanding technical requirements.

Kind regards,

Andre van Wijnen, PhD

Additional Editor Comments (optional):

Editorial Comments: The authors have addressed the final very minor comments from Reviewer#2 and the manuscript does not need to go though a fourth round of review.
---

## [Editor Report · Acceptance letter]

16 Dec 2022

PONE-D-22-10694R3 

Endogenous production of hyaluronan, PRG4, and cytokines is sensitive to cyclic loading in synoviocytes 

Dear Dr. Chan:

I'm pleased to inform you that your manuscript has been deemed suitable for publication in PLOS ONE. Congratulations! Your manuscript is now with our production department. 

Kind regards, 

on behalf of

Dr. Andre van Wijnen 

Academic Editor

PLOS ONE